# Identification of the Differentially Expressed Genes of Muscle Growth and Intramuscular Fat Metabolism in the Development Stage of Yellow Broilers

**DOI:** 10.3390/genes11030244

**Published:** 2020-02-26

**Authors:** Dongfeng Li, Zaixu Pan, Kun Zhang, Minli Yu, Debing Yu, Yinglin Lu, Jiantao Wang, Jin Zhang, Kangning Zhang, Wenxing Du

**Affiliations:** 1College of Animal Science and Technology, Nanjing Agricultural University, Nanjing 210095, China; lidongfeng@njau.edu.cn (D.L.); zaixu_pan@yeah.net (Z.P.); mekge2@yeah.net (K.Z.); yuminli@njau.edu.cn (M.Y.); yunshib071@21cn.com (D.Y.); luyinglin@njau.edu.cn (Y.L.); 2Animal Husbandry Station, Agriculture and Animal Husbandry Bureau of Tangshan Ctiy, Tangshan 063000, China; jinchasx836@21cn.com; 3Jiangsu Lihua Animal Husbandry Co., Ltd., Changzhou 213168, China; mriivx957@yeah.net (J.Z.); zhangkangming0801@163.com (K.Z.)

**Keywords:** chicken, muscle development, gene expression profiling, molecular regulation, fat metabolism

## Abstract

High-quality chicken meat is an important source of animal protein for humans. Gene expression profiles in breast muscle tissue were determined, aiming to explore the common regulatory genes relevant to muscle and intramuscular fat (IMF) during the developmental stage in chickens. Results show that breast muscle weight (BMW), breast meat percentage (BMP, %), and IMF (%) continuously increased with development. A total of 256 common differentially expressed genes (DEGs) during the developmental stage were screened. Among them, some genes related to muscle fiber hypertrophy were upregulated (e.g., *CSRP3, LMOD2, MUSTN1, MYBPC1*), but others (e.g., *ACTC1, MYL1, MYL4*) were downregulated from Week 3 to Week 18. During this period, expression of some DEGs related to the cells cycle (e.g., *CCNB3*, *CCNE2*, *CDC20*, *MCM2*) changed in a way that genetically suggests possible inhibitory regulation on cells number. In addition, DEGs associated with energy metabolism (e.g., *ACOT9*, *CETP*, *LPIN1*, *DGAT2*, *RBP7*, *FBP1*, *PHKA1*) were found to regulate IMF deposition. Our data identified and provide new insights into the common regulatory genes related to muscle growth, cell proliferation, and energy metabolism at the developmental stage in chickens.

## 1. Introduction

Chickens are an important source of high-quality animal protein for humans. The broiler industry is one of the most active industry sectors today, aiming to obtain the highest meat tissue, the optimal ratio of carcass muscle to fat, acceptable physicochemical characteristics, flavor, health, and safety for consumers. Among them, muscle and intramuscular fat (IMF) are the two main characteristics, respectively, representing the yield and quality of meat [1,2]. Essentially, this depends on the development of muscle and adipose tissue or cells, and their relationship.

Muscle fiber is the basic unit of skeletal muscle. It is formed by the fusion of several muscle cells [3]. Myoblasts originate from the mesoderm of the embryo. They proliferate in large numbers, migrate, and fuse into polynucleated cells to form muscle tubes, and then differentiate to form muscle fibers by regulation of Wnt, Shh, MyoD, and Myf5 [4,5,6]. The number of skeletal muscle fibers is mainly determined at the embryonic stage, and the increase of meat produced is mainly due to the increase in muscle cells volume [7,8]. Under the regulation of extracellular factors, new skeletal muscle originates after birth primarily due to the activation, proliferation, and differentiation of satellite cells [9,10].

Energy metabolism plays an important role in muscle growth and development. Skeletal muscle could absorb and utilize glucose and fatty acids, whereas a lack of energy will trigger muscle atrophy, governed by specific signaling pathways. Protein synthesis consumes energy, and a key sensor for cellular energy levels (AMPK) is sensitive to the body’s nutritional status [11]. When the body is deficient in energy and nutrition, skeletal muscle protein synthesis will decrease [12].

Although there are a few studies on the regulatory mechanisms of muscle development and lipid metabolism in chickens [13], knowledge on the common key genes regulating these processes at the developmental stage is scarce. In this study, we focused on exploring the genetic regulation of muscle development and energy metabolism at the developmental stage in chickens. Gene expression profiling was used to identify candidate genes that potentially govern muscle development and lipid metabolism during development. Our findings constitute a theoretical basis for producing higher quality chicken meat.

## 2. Materials and Methods

### 2.1. Animals and Ethics Statement

The study was conducted in accordance with the guidelines for experimental animals developed by the Ministry of Science and Technology of China. The protocols of animal experiments were approved by the Science Research Department (in charge of animal welfare issues) of Nanjing Agricultural University (Nanjing, China; No. NJAU20181102).

The Beijing-You (BJ-Y) chicken, a unique Chinese commercial breed with a high meat quality, was used in this study. One hundred and twenty male BJ-Y chickens with a similar weight at Day 1 came from the same half-sib family and were randomly distributed into four groups. Birds were maintained in 24 floor pens (each 4.55 m^2^) in an environmentally controlled room, at a temperature range of 20–25 °C and relative humidity (RH) between 40% and 70%, throughout the feeding process. Feed and water were provided ad libitum during the experiment. Diets were formulated based on the National Research Council (1994) requirements and the Feeding Standards of Chickens established by the Ministry of Agriculture, Beijing, China (2004). Composition of the diet is shown in Table 1.

### 2.2. Tissue Samples and Measurements

Under carbon dioxide anesthesia, chickens with a similar weight were euthanized by severing the carotid artery at Weeks 3, 8, 13, and 18, respectively (*n* = 20 per time point, five from each group). After slaughter, the pectoral muscles were dissected in the same area for all chickens, snap-frozen in liquid nitrogen, and stored at −80 °C until use for RNA-sequencing. Samples from the pectoral muscle on the other side were stored at −20 °C for biochemical analysis. In addition, the breast meat weight (BMW) and eviscerated weight (EW) were recorded, and breast meat percentage (BMP, %) was calculated (BMW as a percentage of EW).

Two grams of each sample were thawed, obvious fat was removed, and the samples were minced thoroughly. Minced samples were dried in two 10–12-h stages (at 65 °C and 105 °C, respectively), followed by cooling and drying in a desiccator for at least 30 min. The IMF contents in the pectoralis major were measured by the Soxhlet method [14,15], using anhydrous ether as the solvent. Results are expressed as percentages, on the basis of dry tissue weight.

Samples (~2 cm^3^) of 3 randomly selected birds were removed from the same locations on the breast muscle. The samples were oriented for transverse fiber sectioning and mounted on cork disks using OCT Tissue-Tek (Sakura Finetechnical Co., Tokyo, Japan). Serial cryostat sections (10-μm; −20 °C) were cut, mounted, and stained with hematoxylin and eosin [16]. For each bird, muscle fiber size was estimated by measuring the minimum fiber diameter of 100 fibers using image analysis software, and the density of muscle fibers (fibers/mm^2^) was estimated by point-counting stereology, counting 500 points.

### 2.3. RNA Extraction and Gene Expression Profiling

Total RNA was extracted from the pectoral tissue of the chickens at different time points (Weeks 3, 8, 13, and 18), using the TRIzol reagent ((Invitrogen, Carlsbad, CA, USA) according to the manufacturer’s protocol. The quality of the RNA was assessed by 1% gel electrophoresis, and the RNA concentration was determined by a NanoDrop 2000 spectrophotometer (Thermo Fisher Scientific, Hudson, DE, USA). The optical density (OD) 260/280 values of all samples were limited to a range of 1.8 to 2.0. RNA samples were subsequently used for gene expression profiling.

RNA from three representative chickens per week of sampling were selected for transcript detection. Based on ultra-high-throughput sequencing (HiSeq2500; Illumina, San Diego, CA, USA), gene expression profiling was performed at Berry Genomics (Beijing, China). Raw data were converted to FASTQ files using bcl2fastq (Illumina). Clean reads were generated by removing reads containing adapter and low-quality sequences. The results were mapped to the reference chicken genome and genes (*Gallus gallus*, Galgal5; available at https://www.ncbi.nlm.nih.gov/assembly/GCF_000002315.3) using TopHat 1.3.2 (https://ccb.jhu.edu/software/tophat). Gene expression levels were calculated using the RPKM method, as described by Mortazavi et al. [17]. Differentially expressed genes (DEGs) in different time-points comparisons (3w vs. 8w, 3w vs. 13w, and 3w vs. 18w) were analyzed using the R package edgeR. These genes were screened by the following criteria: |log2 FC| ≥  0.58, with *p* < 0.05.

### 2.4. Data Analysis and qRT-PCR Detection

Based on the DEGs, Gene Ontology (GO) enrichment analysis was performed to identify the gene function classes and categories corresponding to the DEGs, using the ClueGO and CluePedia plugins of Cytoscape (https://cytoscape.org/). The significance level of GO terms enrichment was set at *p* < 0.05 as indicated in the Yekutieli method [18]. According to the results of the GO enrichment analysis, the related DEGs were screened. Significantly enriched signaling pathways of DEGs were analyzed by the KEGG (Kyoto Encyclopedia of Genes and Genomes), using Kobas 3.0 [19]. A *p* < 0.05 was considered to be indicative of statistical significance.

Using nine RNA samples from every groups, quantitative real-time polymerase chain reaction (qRT-PCR) was performed to confirm the results of the gene expression profiling. RNA samples were reverse transcribed using a TIANGEN^®^ FastQuant RT Kit (TIANGEN, Beijing, China), and specific primers were designed and shown in Table 2, placing them at or just outside of the exon/exon junctions. Samples were amplified using the real-time PCR Detection System ABI 7500 (Applied Biosystems, Shanghai, China). The PCR mixture contained 10 μL of 2× iQ™ SYBR Green Supermix, 0.5 µL (10 µmol/L) of primers, and 1 μL of cDNA, along with ddH_2_O for a total volume of 20 μL. After initial denaturation for 30 s at 95 °C, amplification was performed for 40 cycles (95 °C for 5 s and 60 °C for 32 s). PCR efficiency for these genes and β-actin was consistent. The comparative cycle threshold (CT) method was used to determine fold-changes in gene expression [20], with fold-changes being calculated as 2^−ΔΔCT^. The results are expressed as the mean fold-change in gene expression from triplicate analyses, using samples of 3-week-old chickens as the calibrators (arbitrarily assigned an expression level of 1 for each gene). Correlations between relative abundance from qRT-PCR and gene expression profiling data were also calculated.

### 2.5. Statistical Analysis

Statistical differences between pairs of groups (3w vs. 8w, 3w vs. 13w, and 3w vs. 18w) were evaluated using the Student’s *t*-test. All computations were performed, using SPSS Version 20.0 (IBM Corporation, Armonk, NY, USA). The Spearman rank correlation analysis was performed to assess the association between data from gene expression profiling and qRT-PCR. A *p* < 0.05 was considered significant, and data are presented as mean ± SEM.

## 3. Results

### 3.1. Changes in Live Weight, Pectoral Muscle, and IMF

Data on the BMW, BMP, and IMF in breast muscle tissue of the chickens at 3, 8, 13, and 18 weeks are presented in Figure 1a. Both of the BMW and BMP (%) have continuously increased (*p* < 0.01) through development from 3 weeks to 18 weeks. Similar observations were recorded for IMF (%), which also continuously and significantly increased (*p* < 0.05 or *p* < 0.01) throughout the development period. In addition, the density and diameter of the muscle fibers were also analyzed, showing that the diameters of the breast muscle fibers continuously and significantly increased, while the density of the fibers accordingly decreased through development from 3 weeks to 18 weeks (*p* < 0.05 for both; Table 3 and Figure 1b).

### 3.2. Identification of DEGs

Using gene expression profiling, a total of 256 commonly known DEGs from three comparisons (3w vs. 8w, 3w vs. 13w, and 3w vs. 18w) were screened. Of these, 86 were downregulated and 170 were upregulated (Appendix A). Gene Ontology (GO) analysis was performed on these 256 DEGs, with the main GO terms being positive regulation of angiogenesis, positive regulation of DNA binding, cell division, kinetochore organization, negative regulation of transcription, cell wall macromolecule catabolic process, regulation of cardiac muscle contraction, muscle contraction, and more (Appendix A). Similarly, using KEGG pathway analysis on these 256 DEGs, eight significantly enriched pathways were found (Appendix A and Figure 2), including pathways related to cell number (apoptosis, cell cycle, and oocyte meiosis). In addition, the focal adhesion, ECM-receptor interaction, glutathione metabolism, phagosome, and pyrimidine metabolism were also screened out.

According to the GO terms and KEGG pathways analyses results, a total of 24 DEGs related to muscle development (*n* = 7), cell number (*n* = 10, including apoptosis, *n* = 4, and cell cycle, *n* = 6), and energy metabolism (*n* = 7, including lipid metabolism, *n* = 5, and glycol-metabolism, *n* = 2) were respectively indicated (Table 4). Moreover, qRT-PCR was performed for these 24 genes to validate the accuracy of the gene expression profiling data, and the association between data from gene expression profiling and qRT-PCR was analyzed by Spearman rank correlation. Results showed that the fold-change of gene expression between the two methods was significantly correlated (*r* = 0.9683, *p* < 0.01; Figure 3).

### 3.3. Differentially Expressed Genes Related to Muscle Development, Cell Number, and Energy Metabolism

As shown in Table 4, the seven common DEGs related to muscle development, at the timeframe studied, were screened and the gene expression fold-change ranged between 2 and 32. Samples from Week 3 were used as control. Results from the qRT-PCR analysis showed that the expression levels of *CSRP3, LMOD2, MUSTN1*, and *MYBPC1* were significantly higher (*p* < 0.05 or *p* < 0.01) at all other time points when compared to those at Week 3, but the expression levels of *ACTC1, MYL1*, and *MYL4* were significantly lower (*p* < 0.01) in the same comparisons (Figure 4). These seven genes were only enriched in the GO terms, such as regulation of cardiac muscle contraction and muscle contraction, but not in the related signaling pathway.

Similarly, seven common DEGs related to glycol-metabolism or lipid metabolism at the developmental stage were screened. Again, samples from Week 3 were used as control. The fold-change found ranged between 2 and 5. qRT-PCR results showed that the expression levels of *ACOT9, CETP*, and *LPIN1* were significantly higher and those of *DGAT2, RBP7, FBP1*, and *PHKA1* were significantly lower at all evaluation time points when compared to those at Week 3 (*p* < 0.01; Figure 5). These genes were found to be mainly involved in the corresponding signaling pathways, but not significantly enriched.

In addition, 10 common DEGs related to cell cycle or apoptosis at the developmental stage were screened. All were shown to have been downregulated when compared to samples from Week 3 that acted as the control. Fold-change ranged between 2 and 11. Using qRT-PCR, we found that the expression levels of these 10 genes, which were enriched in the cell cycle and oocyte meiosis pathways (*BUB1, CCNB3, CCNE2, CDC20, MCM2, PLK1*) or the apoptosis pathway (*BIRC5, CTSK, LMNB1, LMNB2*) (Figure 6), were significantly lower (*p* < 0.05 or *p* < 0.01) at other stages of development when compared to those at Week 3.

## 4. Discussion

Muscle development of broilers closely affects the quantity of chicken products available for human consumption. Intramuscular fat is an important factor affecting meat quality [21,22,23]. Local Chinese chickens have a high meat quality as they have a high IMF content [24]. The developmental stage is the main muscle and IMF formation period. At the same time, myocytes and adipocytes mutually influence each-other due to their close adjacent relationship in the muscle tissues. Identification of co-expressed genes in the muscle tissues during the developmental stage has great significance for controlling muscle production and regulating meat quality. Such analysis could reveal the molecular regulation relationship between them. The remaining energy from the yolk sac can supply the body’s needs for yellow-feather broilers during the first two weeks after hatching [25]. Therefore, four different stages of development were assessed: Week 3 (starting time, acted as baseline control), Week 8 (rapid development), Week 13 (development peak), and Week 18 (market time). Comparisons between Week 3 and the other three time points was performed to screen for common DEGs, which might be the key functional genes affecting muscle or IMF development during the entire developmental period. In addition, the RPKM method was used in the calculation of gene expression levels to obtain more information on the related genes, though the TPM will be more reliable. Meanwhile, the verification of the screened candidate genes was also strengthened by Q-PCR to ensure the reliability of data.

For the 256 DEGs screened out in breast muscle tissue samples at different developmental stages, we further identified the related functional genes by GO and KEGG analyses. Both final muscle and IMF production are the joint result of cell proliferation and differentiation, and energy metabolism plays an important role in this process [26]. Consequently, the functional genes related to muscle development (*n* = 7), cell number (*n* = 10, cell proliferation and apoptosis combined), and energy metabolism (*n* = 7, glucose and lipid metabolism combined) were combed out. Subsequently, these 24 genes were assessed by qRT-PCR in breast muscle tissue, at the different time points, to verify the gene expression profiles data. The verification ratio reached 7.8%, and the correlation analysis showed a high degree consistency between the two methods (*r* = 0.9683, *p* < 0.01), which supports the accuracy of the data from the gene expression profiles.

The number of skeletal muscle fibers is mainly determined at the embryonic stage, and the occurrence of changes in the skeletal muscle after birth is mainly due to the fusion of the muscle satellite cells with the muscle fibers, resulting in hypertrophy of the skeletal muscle fibers [9,10]. In this study, seven common DEGs related to muscle development during the developmental stage were screened out. Among them, *CSRP3, LMOD2, MUSTN1*, and *MYBPC1* levels were significantly higher, and levels of *ACTC1, MYL1*, and *MYL4* were significantly lower at all time points, when compared to Week 3. In a similar study, *MYBPC1* had been shown to have a significantly higher expression level during development in breast muscle tissue [13]. It had been reported that CSRP3, LMOD2, MUSTN1, and MYBPC1 have important positive regulation on myofibril assembly and hypertrophy of muscle fibers [27,28,29], while MYL1 and MYL4 have a negative regulatory effect on myogenesis by inhibiting myoblast proliferation [30]. Phenotypic results showed that BMW, BMP (%), and muscle fiber diameter have all continuously and significantly increased, indicating that muscle growth mainly depends on regulation of the differentiation (hypertrophy) of muscle fiber from 3 weeks to 18 weeks. Comprehensively considering these results, these seven genes were identified as key regulatory genes related to muscle growth at the developmental stage in chickens.

Energy metabolism plays an important role in muscle development. A lack of energy will trigger muscle atrophy by various signaling pathways. It was found that *FBP1* and *PHKA1* mRNA levels have significantly decreased during development. According to published information, FBP1 and PHKA1 play an important regulatory role in gluconeogenesis or glycogen synthesis [31,32]. These results point to the possibility that positive regulation of glucose utilization might be enhanced, and regulation of gluconeogenesis or glycogen synthesis would consequently be reduced in muscle tissue at the developmental stage. For genes related to lipid metabolism, it is known that DGAT2 and RBP7 have an important positive regulatory role in lipid deposition [33,34], ACOT9 and LPIN1 promote lipolysis [35,36], and CETP is involved in reversed cholesterol transport and reduced fat accumulation in chickens [37]. In this study, mRNA expression levels of *DGAT2* and *RBP7* were significantly lower, and those of *ACOT9, CETP*, and *LPIN1* were significantly higher in breast muscle tissue throughout the development period. Combined with the continuous increase in IMF over time, although the rate of increase slowed down gradually, our results suggest that these lipid metabolism-related genes have a regulatory function on IMF deposition at the developmental stage in chickens.

Tissue development is the combined result of cell proliferation and differentiation. Thus, regulation of cell number was also analyzed. It is known that both cell proliferation and apoptosis could affect cell number. These were therefore also a focus of this study. Ten common DEGs related to cell proliferation (*BUB1, CCNB3, CCNE2, CDC20, MCM2,* and *PLK1*, which were mainly enriched in the cell cycle and oocyte meiosis pathways) and apoptosis (*BIRC5, CTSK, LMNB1,* and *LMNB2)* were screened out, and mRNA expression levels of all were significantly lower in breast muscle tissue throughout the development stage. As is widely known, *BUB1, CCNB3, CCNE2, CDC20, MCM2,* and *PLK1* genes have a positive regulatory effect on the cell cycle [38,39,40,41,42], and *BIRC5, CTSK, LMNB1,* and *LMNB2* genes promote cell apoptosis [43,44,45]. The number of skeletal muscle fibers is mainly determined at the embryonic stage, while occurrence of new skeletal muscle after birth is mainly due to activation, proliferation, and differentiation of satellite cells [9,10]. Results on the density and diameter of muscle fibers showed no increase in the number of muscle fibers. Genetically, it was made clear that these 10 key genes regulate cell number in breast muscle tissue at the developmental stage in chickens.

The approach of the present study was to use gene expression profiling to identify the common functional genes that regulate muscle development, IMF accumulation, and cell number in muscle tissue of chickens during development. Possible regulation by translational mechanisms and post-translational modifications may have also contributed. Because of tissue complexity, additional experiments on the expression, localization, and function of the regulatory genes should be further performed to reveal the clear function and regulatory mechanism of these candidate genes on muscle development in chickens.

## 5. Conclusions

In this study, we screened for common regulatory genes related to muscle development, cell number, or energy metabolism at the developmental stage in chickens. Upregulation of *CSRP3, LMOD2, MUSTN1, MYBPC1*, and *MYCBP2*, and downregulation of *ACTC1, MYL1*, and *MYL4,* are associated with muscle development, primarily hypertrophy of muscle fibers, during development from 3 weeks to 18 weeks. Expression change for genes related to lipid metabolism (*ACOT9, CETP, LPIN1, DGAT2*, and RBP7) and glycol-metabolism (*FBP1* and *PHKA1*) may have contributed to the continuous increase in IMF deposition. Meanwhile, the screened-out cell-cycle- and apoptosis-related genes reflect on the negative regulation of cell number at the developmental stage. These findings provide new insights into the regulation of muscle development in chickens.

## Figures and Tables

**Figure 1 genes-11-00244-f001:**
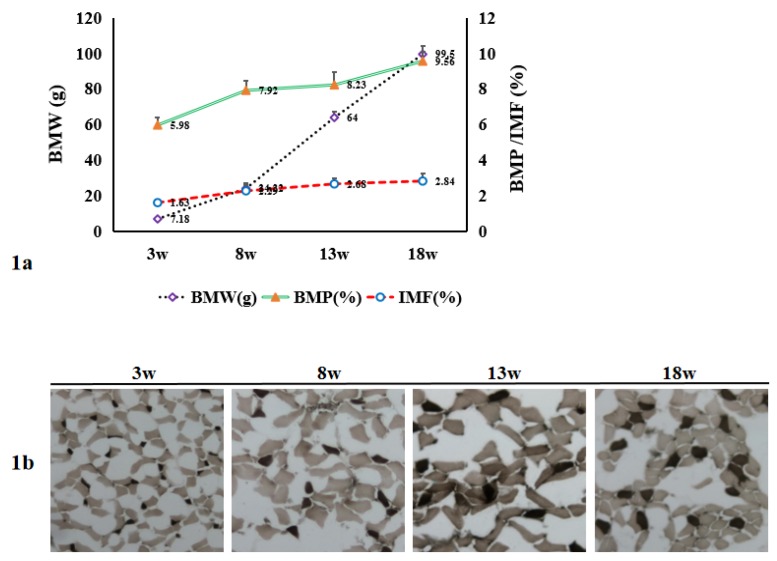
Breast muscle tissue characteristics in male chickens through development from 3 to 18 weeks. (**a**) Breast muscle weight (BMW), breast meat percentage (BMP, %), and intramuscular fat (IMF, %) continuously increased. *n* = 20. (**b**) Continuous hypertrophy of muscle fibers through development. Shown micrographs are at magnification of 40×. *n* = 3.

**Figure 2 genes-11-00244-f002:**
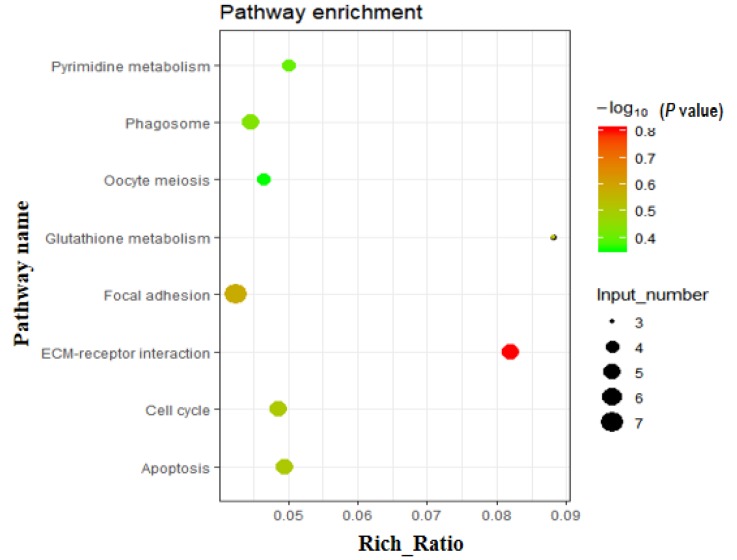
Enriched pathways based on the 256 DEGs. KEGG (Kyoto Encyclopedia of Genes and Genomes) pathway analysis of DEGs showed that various fat metabolism pathways (glycerolipid metabolism, steroid biosynthesis, etc.) were enriched (*p* < 0.05).

**Figure 3 genes-11-00244-f003:**
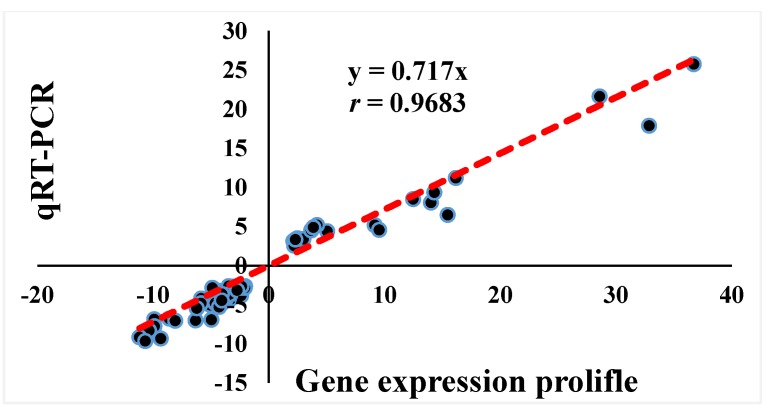
Spearman rank correlation analysis of gene expression profiling and qRT-PCR results of 24 genes in three comparisons. A high correlation coefficient (*r* = 0.9683, *p* < 0.01) was present, indicating that the gene expression profiling data were reliable. *n* = 72.

**Figure 4 genes-11-00244-f004:**
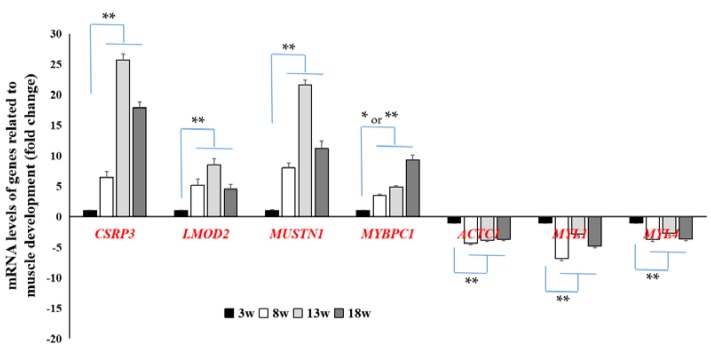
Verification of DEGs related to muscle development by quantitative real-time polymerase chain reaction (qRT-PCR). These DEGs were significantly upregulated or downregulated (**p* < 0.05 or ***p* < 0.01) in breast muscle tissue at 8, 13, and 18 weeks when compared to those at 3 weeks. *n* = 9.

**Figure 5 genes-11-00244-f005:**
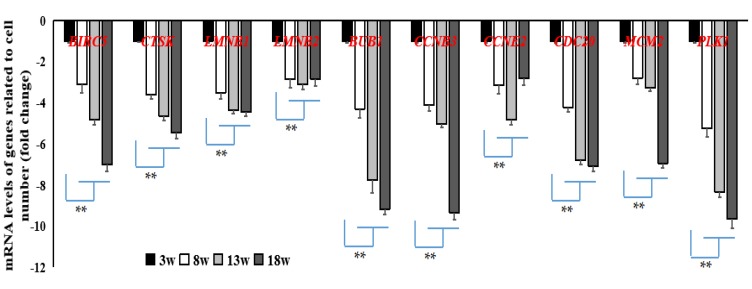
Verification of DEGs related to cell number by quantitative real-time polymerase chain reaction (qRT-PCR). These DEGs were significantly downregulated (***p* < 0.01) in breast muscle tissue at 8, 13, and 18 weeks when compared to those at 3 weeks. *n* = 9.

**Figure 6 genes-11-00244-f006:**
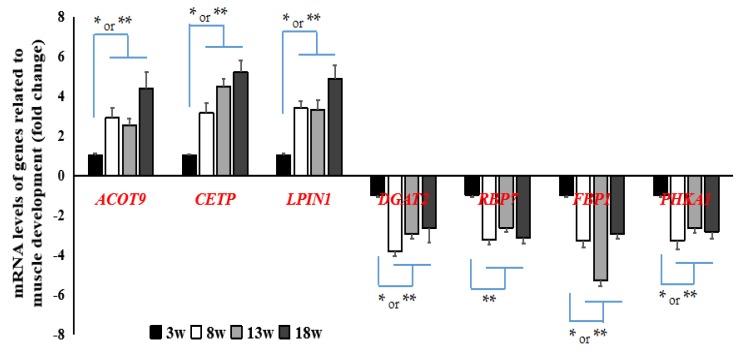
Verification of DEGs related to energy metabolism by quantitative real-time polymerase chain reaction (qRT-PCR). These DEGs were significantly upregulated or downregulated or (**p* < 0.05 or ***p* < 0.01) in breast muscle tissue at 8, 13, and 18 weeks when compared to those at 3 weeks. *n* = 9.

**Table 1 genes-11-00244-t001:** Composition and nutrient levels of experimental diets (% as fed-basis, 22 day–126 day).

Ingredient, %		Nutrient Composition	
Corn	60	ME (kcal/kg)	13.0
Wheat middling	6.85	CP,%	18
Wheat bran	6.6	Ca,%	0.8
Fish meal	0.9	Total P,%	0.6
Feather meal	4.65	Nonphytate P,%	0.35
Soybean meal	12.6	Lys,%	0.85
Lard	4.5	Met,%	0.32
L-lysine HCl	0.2	Met+Cys,%	0.69
DL-Methionine	0.09	Thr,%	0.66
Limestone	1.17	Trp,%	0.17
Dicalcium phosphate	1.14	Ile,%	0.68
Salt	0.3		
Premix ^1^	1.00		
Total	100.00		

^1^ Provided the following per kilogram of diet: retinyl acetate, 10,000 IU; cholecalciferol, 2000 IU; DL-α-tocopherol acetate, 20 IU; menadione, 2.50 mg; thiamine, 2 mg; riboflavin, 8 mg; niacin, 50 mg; pyridoxine, 8 mg; cobalamin, 0.01 mg; pantothenic acid, 20 mg; folic acid 0.8 mg; biotin, 0.18 mg; choline chloride, 500 mg; Fe, 80 mg; Cu, 8 mg; Mn, 80 mg; Zn, 60 mg; I, 0.35 mg; Se, 0.15 mg.

**Table 2 genes-11-00244-t002:** The specific primers for Q-PCR in this study.

Gene	Sequence	Accession NO.
*CSRP3*	F:5′-CTTTGGACAAGGGGCTGGAT-3′ R:5′-TCTGCAGCGTACACCGATTT-3′	NM_001199486
*LMOD2*	F:5′-GGGTGCGTGTGAGAAGGATT-3′ R:5′-CTGGAACTCCTGCCATCCTC-3′	NM_001199715
*MUSTN1*	F:5′-CCCTTGCACTAAGCTCACCA-3′ R:5′-ACGTAGAAAGAAGGCCCGTG-3′	NM_213580
*MYBPC1*	F:5′-CACGGTGGATGAGGCTGAAT-3′ R:5′-CTGCTCCAATGTGGTCTGGT-3′	XM_025155757
*ACTC1*	F:5′-CCGTGCCTATCAGCCAAGAT-3′ R:5′-CGACGATGGATGGGAACACA-3′	NM_001079481
*MYL1*	F:5′-TCGGAAAGACCAGATGGCAC-3′ R:5′-TTTCCACAACCCCCGTGAAA-3′	NM_001044632
*MYL4*	F:5′-TCAAGAAACCCGACCCCAAG-3′ R:5′-CGTAGGTGATCTGCATGGCT-3′	NM_205479
*BIRC5*	F:5′-GCCTATGCTGAAATGCTGCC-3′ R:5′-CGCGGAGTGCTTTTTGTGTT-3′	NM_001012318
*CTSK*	F:5′-CCGCCATAAAAGAGCCAACG-3′ R:5′-GTCCTCTTCCAGAGGTCCCA-3′	NM_204971
*LMNB1*	F:5′-AGGAGCGGGAAAACTATCGC-3′ R:5′-ACTACGGCTTGACGAAGCTC-3′	NM_205286
*LMNB2*	F:5′-ACTTATGCGTGTGGACCTGG-3′ R:5′-CCGACTGGTGTCCACTTCAA-3′	NM_205285
*BUB1*	F:5′-AAGTTACGAGGCGCAGATCC-3′ R:5′-GTCACGAACGCCTTCACAAG-3′	NM_001012870
*CCNB3*	F:5′-GCTACTTTCAAAAGAGCCGGG-3′ R:5′-AACGCTGACCTCTTCTTGGG-3′	NM_205239
*CCNE2*	F:5′-GATGTCGAGACGCAGCCGA-3′ R:5′-TTCTTCTTAATCTCCTCTGCCGTT-3′	NM_001030945
*CDC20*	F:5′-ATTCCCAACCGCAGCACTAT-3′ R:5′-AGCAGGTGTAGTCTTCTGGC-3′	NM_001006536
*MCM2*	F:5′-TAATCCGGCGGGGTAGGAA-3′ R:5′-GTAGTCCCTCTCCATCCCCT-3′	NM_001006139
*PLK1*	F:5′-TCATCCTGGGCTGCCAATAC-3′ R:5′-TCTTGGGCTCGCCATCATAC-3′	NM_001030639
*FBP1*	F:5′-AATCTTGTGGCAGCGGGTTA-3′ R:5′-CTGCCGTCCTCAGGGAATTT-3′	NM_001278048
*PHKA1*	F:5′-AGAAGAGTGTGCGATCGGTG-3′ R:5′-GGTCAGAGACTGCCTACGTT-3′	XM_004940598
*ACOT9*	F:5′-CTATGGTGCTGGAGGACCGC-3′ R:5′-CCTCAATCTGCTCCGCACTT-3′	NM_001012823
*CETP*	F:5′-AGTCTCGCCCTTCCTGAGAT-3′ R:5′-GCAGCTTGGATAGTGACCGT-3′	NM_001034814
*LPIN1*	F:5′-ACCATGGCAAACAGAATAAAAGATG-3′ R:5′-CCTTCACAGCGGCAAGTACC-3′	XM_015276093
*DGAT2*	F:5′-ATGGGTCCTCACGTTCCTCA-3′ R:5′-CCACTGGGATCTTCTTCCACC-3′	XM_419374
*RBP7*	F:5′-GAAGAACAGGGGCTGGACTC-3′ R:5′-TGCATGGCTGTCATGTTTCC-3′	XM_417606
*β-actin*	F:5′TCTTGGGTATGGAGTCCTG-3′ R:5′TAGAAGCATTTGCGGTGG-3′	NM_205518

**Table 3 genes-11-00244-t003:** The diameter and density of muscle fibers in breast tissue at different stages of development.

Stage (weeks)	Muscle Fiber Diameter (μm)	Muscle Fiber Density (fibers/mm^2^)
3	1.85 ± 0.24 ^a^	1971.26 ± 87.84 ^a^
8	5.61 ± 0.51 ^b^	1448.51 ± 60.22 ^b^
13	16.93 ± 1.19 ^c^	1089.36 ± 44.82 ^c^
18	25.77 ± 1.07 ^d^	882.14 ± 55.33 ^d^

^a,b,c,d^ Means within a column with different superscripts differ significantly (*p* < 0.05). *n* = 3.

**Table 4 genes-11-00244-t004:** The screened 24 DEGs related to muscle development, cell number, and energy metabolism from data of gene expression profiling.

Terms	Ensemble	Gene	Fold Change			Regulation
			3w vs. 8w	3w vs. 13w	3w vs. 18w	
Muscle development	ENSGALG00000004044	*CSRP3*	15.473385	36.73958	32.871784	down
	ENSGALG00000008805	*LMOD2*	9.161582	12.490921	9.54745	down
	ENSGALG00000001709	*MUSTN1*	14.036668	28.596878	16.179874	down
	ENSGALG00000012783	*MYBPC1*	2.4964237	3.8927796	14.324914	down
	ENSGALG00000009844	*ACTC1*	3.3218217	3.7541575	3.3806047	up
	ENSGALG00000002907	*MYL1*	9.879383	4.85955	5.83294	up
	ENSGALG00000000585	*MYL4*	4.1020255	3.218896	4.0335712	up
Apoptosis	ENSGALG00000008713	*BIRC5*	2.5183046	4.0182652	6.315548	up
	ENSGALG00000028147	*CTSK*	3.096479	4.440422	6.216727	up
	ENSGALG00000014692	*LMNB1*	3.006749	4.260736	4.046107	up
	ENSGALG00000000470	*LMNB2*	3.2512372	2.5945606	2.236566	up
Cell cycle	ENSGALG00000008233	*BUB1*	3.6982634	9.865544	11.166059	up
	ENSGALG00000025810	*CCNB3*	4.101872	5.044319	9.328714	up
	ENSGALG00000040794	*CCNE2*	2.4160562	4.6105256	2.6012228	up
	ENSGALG00000009971	*CDC20*	5.8222446	8.590813	8.071098	up
	ENSGALG00000006037	*MCM2*	2.4000816	2.9283285	4.938962	up
	ENSGALG00000006110	*PLK1*	4.4251847	10.326489	10.658555	up
Glycometabolism	ENSGALG00000012613	*FBP1*	2.3082318	4.2890453	2.5612748	up
	ENSGALG00000004801	*PHKA1*	2.8975124	2.043697	2.0698583	up
Lipid metabolism	ENSGALG00000016351	*ACOT9*	2.238887	2.2152865	5.0676374	down
	ENSGALG00000001234	*CETP*	2.1345193	3.6858957	4.175847	down
	ENSGALG00000016456	*LPIN1*	3.00475	2.3144455	3.8659291	down
	ENSGALG00000040418	*DGAT2*	2.3845758	2.4465158	2.3423936	up
	ENSGALG00000002637	*RBP7*	4.246696	3.4545891	2.7316358	up

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
