# Peer review of "Identification of the Differentially Expressed Genes of Muscle Growth and Intramuscular Fat Metabolism in the Development Stage of Yellow Broilers"

_genes, 2020, doi:10.3390/genes11030244_

Round 1

Reviewer 1 Report

First paragraph lacks references. Can authors explain why male individuals were used in this study? Moreover, choosing the Beijing-You (BJ-Y) chicken as experimental should be better justified. Are the results of this work relevant in the context of anther chicken breeds? Furthermore, the Authors should explain why all experiments were performed at weeks 3, 8, 13, 18. Composition of diet should be also included. In addition more details regarding animals maintaining should be added. Was primer concentration 10 μmol or 10 μmol/l? I also failed to find the list of primers used in this study. Statistical analysis of correlations performed in this work should be descripted. Furthermore, can Authors explain why they used t-test? The Authors studied four groups, thus ANOVA would be more suitable for statistical analysis. Can Authors explain why Fig. 1a lacks SEM values? The list of abbreviations should be included.

Reviewer 2 Report

Comments to the manuscript are listed below.

・How about the pedigree structure of the chickens used in the current study? Also, the authors need to describe about rearing environment in detail, such as temperature and humidity condition.

・Why did the authors use RPKM, but not TPM?

・The authors should also discuss about the individual genes screened in more detail, not only GO and pathway analyses.

・The authors need to discuss why oocyte meiosis and apoptosis pathways were identified in the current study.

・English proofreading is recommended.

Reviewer 3 Report

This study provides a longitudinal gene expression analysis of chicken pectoralis muscle at four time points of postnatal life. Over this time course, 256 commonly differentially expressed (DEG) genes were identified. Amongst these, 24 genes related to muscle development and other processes were selected and corroborated by RT-qPCR.

From this analysis the authors claim “our data reveal the common genes that regulate muscle development, cell number, and energy metabolism at the developmental stage in chickens.” While the few genes identified as differentially regulated in this setup indeed are associated with these processes, the approach taken and the results presented by far do not support such universal claims.

The study provides a dataset of gene expression of postnatal chicken pectoralis muscle. However the authors do not attempt in any way to analyze or demonstrate that the genes they identified are functionally linked in regulating these processes. To me it rather appears that the genes identified in this study are a consequence of the differentiation process and do not allow any speculation towards regulatory processes. The authors attempt constructing such connections, but in my view this is erroneous. E.g. it is stated “MYL1 and MYL4 have negative regulatory effect on muscle development [27-29].” Refs 27, 28 demonstrated timed expression of MYLs upon myogenic differentiation, ref. 29 shows a deleterious effect of precocious expression of a MYL gene upon genetic manipulation. If I am not mistaken, none of the references suggests an intrinsic negative regulatory role for MYLs in myogenesis. The title and claims made in the abstract or introduction are overstatements; there is no data showing any “effect” of the detected DEGs on muscle development or metabolism (title). The identification of 7 DEGs related to energy and lipid metabolism do not allow drawing a conclusion on “…inhibition on the decrease in IMF deposition rate” (abstract), particularly on the basis of bulk tissue analysis (see below).

Additional remarks

Fig. 1B and text referring to it (“density and diameter of muscle fibers were also analyzed, showing that the diameters of breast muscle fibers continuously and significantly increased, while the density of fibers accordingly decreased through development”) suggest quantification; fiber diameter should be quantified (e.g. minimum Feret’s diameter) as well as fiber density.

Fig. 2: what does “Rich_Radio” on the X-axis mean?

A weakness is the approach: RNA Sequecing of bulk tissue. This does not allow for allocation of DEGs to a certain cell type (Fiber? Myoblast? Mesenchymal cell? Adipocyte?). To this end, the authors discussion addressing e.g. intramuscular fat are merely speculative.

The title is seriously overstating the manuscript content; there is no data showing any effect of the detected DEGs on muscle development or metabolism.

Round 2

Reviewer 2 Report

>Question 1: How about the pedigree structure of the chickens used in the current study? Also, the authors need to describe about rearing environment in detail, such as temperature and humidity condition.

Response: Thanks for your question.

As per your suggestion, we had supplemented the pedigree structure of the used chickens, and the details about rearing environment.

Comments to the authors:

The authors need to describe what "the same genetic background" means in more detail.

> Question 2: Why did the authors use RPKM, but not TPM?

Response: Thanks for your reminder.

We have also been concerned that the data analyzed by TPM will be more reliable, but the amount of data gets smaller. Based on this consideration, RPKM was still used for analysis to obtain more information on the related genes. Meanwhile, we also strengthen the verification of the screened candidate genes by Q-PCR to ensure the reliability of data.

Comments to the author:

I think it is better to include the content of response described above in the manuscript.

Question 4: The authors need to discuss why oocyte meiosis and apoptosis pathways were identified in the current study.

Response: Thanks for your question.

As per your suggestion, we had supplemented the discussion on the reason of the identification of oocyte meiosis and apoptosis pathways in discussion section.

Comments to the author:

Sorry, but finally I could not find any new sentences about the discussion on the reason of the identification of oocyte meiosis in discussion section.

Author Response

Dear reviewers,

Thanks for your questions and comments. These have been explained or revised in the manuscript, point-by-point with using yellow highlighting.

Reviewer 2:

>Question 1: How about the pedigree structure of the chickens used in the current study? Also, the authors need to describe about rearing environment in detail, such as temperature and humidity condition.

Response: Thanks for your question.

As per your suggestion, we had supplemented the pedigree structure of the used chickens, and the details about rearing environment.

Comments to the authors:

The authors need to describe what "the same genetic background" means in more detail.

Response: Thanks for your question.

As per your suggestion, we had supplemented this in “Materials and Methods” section of manuscript (Lines 66-67).

> Question 2: Why did the authors use RPKM, but not TPM?

Response: Thanks for your reminder.

We have also been concerned that the data analyzed by TPM will be more reliable, but the amount of data gets smaller. Based on this consideration, RPKM was still used for analysis to obtain more information on the related genes. Meanwhile, we also strengthen the verification of the screened candidate genes by Q-PCR to ensure the reliability of data.

Comments to the author:

I think it is better to include the content of response described above in the manuscript.

 Response: Thanks for your question.

As per your suggestion, we had supplemented this in “Discussion” section of manuscript (Lines 268-271).

Question 4: The authors need to discuss why oocyte meiosis and apoptosis pathways were identified in the current study.

Response: Thanks for your question.

As per your suggestion, we had supplemented the discussion on the reason of the identification of oocyte meiosis and apoptosis pathways in discussion section.

Comments to the author:

Sorry, but finally I could not find any new sentences about the discussion on the reason of the identification of oocyte meiosis in discussion section.

Response: Thanks for your question.

As per your suggestion, we had supplemented and adjusted this in “Discussion” section of manuscript (Lines 314-322).

Reviewer 3 Report

The authors have adequately addressed my concerns by mainly changing the text of the manuscript and thereby attenuating previous overstatements.

Author Response

Dear reviewers,

Thanks for your questions and comments. These have been explained or revised in the manuscript, point-by-point with using yellow highlighting.

Reviewer 3:

The authors have adequately addressed my concerns by mainly changing the text of the manuscript and thereby attenuating previous overstatements.

Response: Thanks for your approval.

We also appreciate your help, which has greatly improved the quality of our manuscript.